# Isolation and Extraction of Monomers from Insoluble Dietary Fiber

**DOI:** 10.3390/foods12132473

**Published:** 2023-06-24

**Authors:** Junyao Wang, Jiarui Zhang, Sainan Wang, Wenhao Liu, Wendan Jing, Hansong Yu

**Affiliations:** 1College of Food Science and Engineering, Jilin Agricultural University, Changchun 130118, China; wjunyaon1@163.com (J.W.); jiarui197@163.com (J.Z.); lwh1766850992@163.com (W.L.); 2National Soybean Industry Technology System Processing Laboratory, Changchun 130118, China

**Keywords:** insoluble dietary fiber, cellulose, hemicellulose, lignin, mono-component modification

## Abstract

Insoluble dietary fiber is a macromolecular polysaccharide aggregate composed of pectin, glycoproteins, lignin, cellulose, and hemicellulose. All agricultural by-products contain significant levels of insoluble dietary fiber. With the recognition of the increasing scarcity of non-renewable energy sources, the conversion of single components of dietary fiber into renewable energy sources and their use has become an ongoing concern. The isolation and extraction of single fractions from insoluble dietary fiber is one of the most important recent research directions. The continuous development of technologies for the separation and extraction of single components is aimed at expanding the use of cellulose, hemicellulose, and lignin for food, industrial, cosmetic, biomedical, and other applications. Here, to expand the use of single components to meet the new needs of future development, separation and extraction methods for single components are summarized, in addition to the prospects of new raw materials in the future.

## 1. Introduction

The most basic description of dietary fiber is given in the GB/Z21922 “Basic Terminology of Food Nutrients” developed by China [1]. The definition of dietary fiber in its statutes is as follows: polymers of carbohydrates that are naturally present in plants, extracted from plants, or directly synthesized with a degree of polymerization ≥ 3; edible yet not digested or absorbed by the human small intestine; and of health significance to humans [2,3]. The “Chinese Dietary Guidelines for Residents 2016” recommends a daily intake of 25–30 g of dietary fiber, which should include grains, potatoes, vegetables, fruits, meat, eggs, dairy products, soybeans, and nuts in one’s daily diet. Dietary fiber can be divided into water-soluble and non-water-soluble forms, and is mainly composed of non-starch polysaccharides from various plant substances, including cellulose, lignin, wax, chitosan, pectin, beta-glucan, inulin, and oligosaccharides [4]. Cellulose, hemicellulose, and lignin are the main components of insoluble dietary fiber, whereas pectin is a form of soluble dietary fiber found in non-fibrous substances, such as barley, legumes, carrots, citrus, flax, oats, and oat bran [5].

Dietary fiber is defined as the “seventh macronutrient” in terms of its physiological function [6]. As the standard of living and awareness increases, many people are consuming dietary fiber, which can increase stool volume, promote intestinal peristalsis, and improve bowel patterns [7]. It can also lower total and LDL cholesterol levels in the blood to prevent coronary heart disease, lower fasting and postprandial blood sugar and insulin levels, and improve insulin sensitivity [8]. Finally, it can provide energy-producing metabolites to the colon, increase the quantity and activity of beneficial bacteria, and suppress obesity [9]. IDF is important as a functional food ingredient with multiple health and nutritional benefits [10]. Dietary fiber is incorporated into bread products to increase their nutritional value and sensory quality [11], and into beverages to compensate for dietary deficiencies to a certain extent [12]. Ginseng-IDF has a typical hydrolyzed fiber structure, polysaccharide functional groups, and cellulose crystal structure, and can also be used as an ideal functional ingredient for food processing [13]. It is also used in the production of meat products to obtain high-dietary fiber, low-calorie, low-fat products; it can make the meat taste richer [14]. Developing economically viable and sustainable technologies for converting corn fiber into liquid fuels is widely seen as a promising approach to achieve improved ethanol titers and integrated utilization of byproducts in the traditional corn dry milling process [15].

Research on and the development of dietary fiber can significantly improve the economic value of agricultural products, which has profound significance for improving the health of the population and the economic efficiency of agriculture. The importance of dietary fiber in people’s lives has been confirmed, and if the insoluble component of dietary fiber can be exploited, it could also be used as a substitute in the material industry, in textiles, and in nano-materials. Therefore, this review aims to provide guiding information using selected literature and experimental data, focusing on the isolation and extraction of cellulose, hemicellulose, and lignin from insoluble dietary fiber, and to discuss techniques for the modification of single components.

## 2. Insoluble Dietary Fiber

Insoluble dietary fiber is a type of non-starch polysaccharide that is insoluble in water and cannot be digested in the small intestine or fermented in the colon, and this includes cellulose, hemicellulose, and lignin [16]. Insoluble dietary fiber, which is the main component of the cell wall, can be extracted using physical, chemical, enzymatic, and combined enzyme–chemical methods. Further, it is completely hydrolyzed to form various monosaccharides [17], including glucose, xylose, galactose, arabinoxylan, and galacturonic acid. In terms of its chemical composition (Figure 1), the chemical structures of the various monosaccharide molecules that comprise the macromolecules are not different. Modified insoluble dietary fiber has a larger specific surface area, higher water-holding and swelling capacities, and greater functionality, based on adding or enhancing functions that were previously absent or weak. However, different binding methods produce insoluble dietary fiber-specific physicochemical properties, which in turn affect physiological functions. Functional groups (such as hydroxyl, carboxyl, amino, and acetyl) in insoluble dietary fiber confer strong hydrophilic, lipophilic, swelling, and metal ion adsorption properties [18]. In addition, these compounds have physical and chemical properties that include thermal stability, rheological properties, ion exchange capacity, and particle distribution. The water-holding and swelling forces are mainly related to the hydrophilic groups on the surface of insoluble dietary fiber and the honeycomb porous structure; its water-holding capacity is approximately 1.5–25 h based on its weight, whereas its oil-holding capacity is mainly related to the hydrophobic zone of fibers and the pore-like structure formed by cross-linking between different components [19]. Insoluble dietary fiber contains side chain groups, such as amino and carboxyl groups, which are reversibly exchanged with heavy metal ions, causing harmful ions to be adsorbed on the fiber and then eliminated in feces. These properties are closely related to the physiological functions of insoluble dietary fiber in the human body. Therefore, there is an urgent need to analyze the monocomponent structures of insoluble dietary fiber.

Cellulose, one of the important components of insoluble dietary fiber, is a polysaccharide polymer consisting of linear chains of D-glucose units linked by β(1→4) glycosidic bonds [20]. Cellulose has a fibrous porous structure and is a commercially important biopolymer with a good physical structure [21]. Its advantages include high crystallinity, high polymerization, a high specific surface area, and good adsorption properties, which offer a great scope for the development of renewable energy. As the most abundant natural polymer, efforts have been made to isolate and extract cellulose for use in new materials [22]; specifically, some of the cellulose is exploited in daily food, decoration, ceramic, paint, tobacco agriculture, explosive, electrical, and construction materials [23]. There is also a portion of fiber that is mainly found in fruits and vegetables, as foods that are consumed by humans. However, modified cellulose is mostly used to adsorb heavy metal ions from wastewater for recycling purposes, owing to its enhanced adsorption capacity [24]. Many companies have separated, purified, and dissolved raw cellulose materials to produce rayon, acetate, sodium carboxymethylcellulose derivatives, methylcellulose, and cellulose membranes.

Unlike cellulose, hemicellulose is a general term used to describe a variety of complex sugars [3]. The basic structural units of hemicellulose are D-arabinose, D-galactose, D-glucose, D-xylose, D-mannose, D-galacturonic acid, D-glucuronic acid, and 4-O-methylglucuronic acid, with small amounts of L-amylose and L-rhamnose [25]. They occur in various structural forms and are divided into four major groups: xylans, mannose, mixed-link-glucans, and xyloglucans [26]. Hemicellulose has a special chemical structure and unique physiological functions. It can be used for fermentation to produce ethanol [27], for microbial screening [28], and as a thermoplastic and water-resistant material [29]. Hemicellulose, one of the main components of insoluble dietary fiber, is abundant, biodegradable, renewable, and biocompatible in nature. Moreover, it is widely used in many fields, such as food packaging materials [30], paper coatings [31], chemicals [32], environmental energy, and biomedicine.

Another important component of insoluble dietary fiber, lignin, is mainly composed of three elements: carbon (C), hydrogen (H), and oxygen (O). It can also contain small amounts of nitrogen (N) and sulfur (S) depending on the source and extraction method [33]. Lignin is tightly complexed with cellulose and hemicellulose, and is a biopolymer with a three-dimensional network structure formed by three benzene propane units interconnected by ether and C-C bonds; moreover, it is rich in aromatic ring structures, aliphatic and aromatic hydroxyl groups, quinone groups, and other reactive groups [34]. Lignin is used to prevent cardiovascular diseases due to its ability to modify the activity of microorganisms in the intestinal system and to lower cholesterol and blood sugar levels. In addition, it has antioxidant activity [29] and possesses functions such as cancer cell-inhibitory activity. In addition to its physiological functions, lignin is used in the production of composite materials because it is abundant, cheap, renewable, degradable, and non-toxic. It is also used as a filler in biomass materials, such as rubber and hydrogel [35].

**Figure 1 foods-12-02473-f001:**
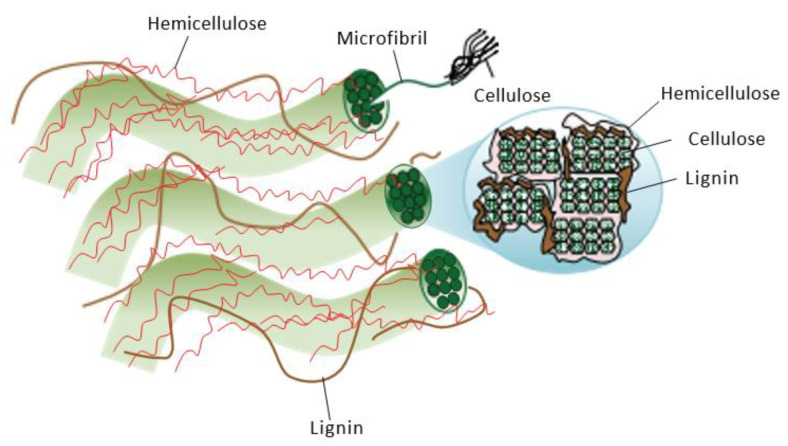
Plant cell wall structure and microfibril cross-section (strands of cellulose molecules embedded in a matrix of hemicellulose and lignin) [36].

## 3. Cellulose

### 3.1. Separation and Extraction Methods

Cellulose is a renewable and degradable biopolymer with high thermal stability. Depending on the separation and extraction method, cellulose has been used in food processing [37], plastic bags, cling films, the textile industry, and the bio-pharmaceutical industry [38]. Different separation and extraction methods are required in order to obtain cellulose from this biopolymer for various applications. This section provides an overview of this, based on various separation and extraction methods.

#### 3.1.1. Acid Hydrolysis Method

The acid hydrolysis separation method involves hydrolysis of the amorphous regions of cellulose in dietary fibers, leaving the compact and dense crystalline regions intact. The rice husk is soaked overnight in 4 wt% NaOH, and the lignin and hemicellulose in the rice husk fibers are removed via acid hydrolysis (H_2_SO_4_). The mixture is then transferred to a round-bottomed flask and refluxed for 2 h (Figure 2). Next, the solid is filtered and washed three times with distilled water, which has been found to result in a cellulose content of approximately 31%. Acid hydrolysis is the most widely used method for preparing cellulose and cellulose-derived cellulose nanocrystals, cellulose filaments, and nanosized cellulose [39]. Through pre-treatment with bagasse, a 50% bagasse cellulose content was obtained after acid hydrolysis at 45 °C for 75 min. Owing to the long acid hydrolysis time, the crystalline structure of cellulose was completely destroyed, resulting in the formation of a needle-like structure [40]. The increase in available energy consumption in recent years has intensified the development of renewable energy sources, especially palm empty fruit bunch fibers, which are present in large quantities and have a high cellulose content [41]. Palm oil empty fruit bundles were then hydrolyzed by adding acid, at a 55% concentration, for 3 h at 45 °C to obtain cellulose with a purity of 14.98% [42]. The production of cellulose with a controlled structure, based on its isolation and extraction from bamboo and representing a dietary fiber of plant origin, has been studied as an important method to obtain good sustainability. Cellulose was successfully extracted by immersing 50 g of treated bamboo powder in a 7.5% NaClO_2_ solution (*w*/*v*) and incubating it at 80 °C for 2 h under acidic conditions (pH = 3.8–4.0). However, the cellulose was found to exhibit different shapes depending on the acid concentration used for hydrolysis [43].

#### 3.1.2. Steam Blasting Method

Steam blasting is a physical separation method that is used for cellulose extraction. The principle behind this method is to break down the cell wall’s structure. The effective heat carrier steam rapidly heats the raw material to a specified temperature; after a period of contact, the steam penetrates the raw material and immediately releases pressure, which results in the degradation of a small portion of cellulose and hemicellulose in the raw material into monosaccharides [45]. Typical steam blasting process temperatures range from 160 to 260 °C, corresponding to pressures of 0.69–4.83 MPa. Agricultural wastes, such as wheat straw and corn straw; specialized energy crops, including manzanita; and willow have been proposed as biomass resources for xylose release to produce xylitol, and the initial xylose hydrolysis products have been found to be released up to 94% under steam blast treatment with 12 Pa for 3 min with 1.2% phosphoric acid and 500 g of substrate [46]. The effects of steam blasting—with citric acid, sodium hydroxide, and water as catalysts—on the chemical properties, structural properties, and enzymatic processes of sugarcane bagasse were investigated. During the citric acid-catalyzed blast treatment, cracks appeared in the fiber cell walls and the maximum hemicellulose removal rate reached 41.5%, whereas in the NaOH-catalyzed steam blast treatment, the bagasse fibers were complete destroyed and the lignin removal rate reached 65%; the water treatment obtained a maximum cellulose yield of 97.5% [47]. By combining the steam blasting process with the traditional scraping method, the upper waterproof layer, which adversely affects the steam blasting process, was removed from the fresh blades and cut for steam blasting treatment, yielding 85.4% cellulose [48].

#### 3.1.3. Deep Eutectic Solvent (DES) Method

The DES comprises mixtures of two or three safe and inexpensive compounds bonded to each other via hydrogen bonding to form salt solutions [49]. Considering food safety and acceptability, cellulose extraction using a DES is the best choice. Choline chloride-glycerol, choline chloride-urea, and choline chloride-oxalic acid were heated and stirred for 2 h at 80 °C, according to a certain molar ratio, to obtain a homogeneous and clarified DES solvent. Then, 2 g of ramie fiber was added to 200 g of DES, the suspension was heated in a closed flask at 100 °C in an oil bath, and the mixture was stirred and mixed at a predetermined temperature for 2–10 h. The purity of the obtained cellulose was 73–78% [50] (Figure 3). Three DES solvents were prepared, and 0.1 g of the sample was mixed with the prepared DES, heated, and stirred in an oil bath at 130 °C for 3 h. The DES prepared with ChCl-LA was the best choice for cellulose extraction. In addition to the well-known sugarcane, straw, and bamboo, in which the dietary fiber content is high, the insoluble dietary fiber component of okara is also rich, with a purity of 90% or higher [51]. The application of DES in agro-industrial treatments can be developed for industrial use. To simplify the defatting, deproteinization, and cellulose extraction steps during the cellulose pretreatment of okara, a new method for cellulose extraction, by preparing a DES solvent via a heating method, was established. Treatment of okara with the DES solution ChCl-O had a significant effect on extracted okara cellulose, improving the thermal stability and cellulose content to 92.6% [52]. To achieve an economic and ecological balance, cheaper methods for cellulose extraction must be explored. A low-eutectic solvent (DES) system of ChCl-EG was used as the solubilizing solvent. DES was used as the reaction medium and mixed with the insoluble dietary fiber of okara at 120 °C for 2 h. The yield of raw cellulose was 84–87% [53].

#### 3.1.4. Comparison of Different Methods

Each method for separating and extracting cellulose has its own advantages and disadvantages. Table 1 presents a detailed visual comparison of the three methods for separating single components, which can provide a theoretical basis for selecting the most suitable method for future experiments.

### 3.2. Modification Technology

Plant cellulose is a natural renewable resource, and its modification is a popular research topic. This section provides a brief overview of the modification methods for cellulose, including physical, chemical, and biological approaches. The most important methods include esterification, sulfonation, etherification, ether esterification, cross-linking, and graft copolymerization. The chemical modification of cellulose refers to the use of coupling agents (e.g., citric acid, malic acid, and tartaric acid) to replace the hydroxyl groups on the surface, thus reducing the content of hydroxyl groups on the cellulose surface and weakening the cellulose and hemicellulose [54]. A series of reactions usually involves a hydroxyl group in the structure. Through modification, a series of ionic groups is introduced which enhances the hydrophilicity of cellulose. The most widely used method is the chemical modification of cellulose, and chemically modified cellulose-based nano-materials are considered one of the best nano-materials [55].

#### 3.2.1. Physical Modification Method

The physical modification of cellulose involves changing its morphology and surface structure via physical and mechanical means, without changing its chemical composition or the chemical reaction. The most commonly used physical method is polyelectrolyte adsorption, which is the only method described in the following section [56], and it can be divided into two categories: polyelectrolytes and nonelectrolytes physically adsorbed on the cellulose surface. Uncharged polymers can bind to cellulose via van der Waals and hydrogen-bonding forces. The modified nanocellulose was firstly adsorbed with the cationic polyelectrolytes poly(DMDAAC), poly(allylamine hydrochloride), and poly(PEI), and, alternatively, the anionic polyelectrolyte poly (4-sodium allyl sulfonate); then, the modified nanocellulose was adsorbed with cationic polyelectrolytes; and finally, the complex material of nanocellulose and nano-silver was prepared by loading silver nanoparticles [57]. The results obtained from studying the antimicrobial activity of this complex material against *Staphylococcus aureus* and *Klebsiella pneumoniae* showed that both the polyelectrolytes and nanosilver were effective; however, the antimicrobial effect of nanosilver was crucial. Finally, it was used as a filler for starch-based coating-modified eucalyptus blue paper, and the application potential of the antibacterial paper products was studied. Thermally responsive nanocellulose was prepared via the adsorption of thermally responsive polyelectrolytes onto nanocellulose. Three block copolymers consisting of quaternized poly(2-dimethylamino) ethyl methacrylate (PDMAEMA) as the polyelectrolyte block and poly(ethylene glycol) methyl ether methacrylate (PDEGMA) as the thermoresponsive block were synthesized [56]. Block copolymers were synthesized through two-atom transfer radical polymerization (ATRP), in which PDMAEMA macromolecular chains were first synthesized as macromolecular initiators for the synthesis of PDEGMA polymers. Among the three block copolymers, the lengths and charges of the PDMAEMA blocks remained the same, whereas the molecular weights of the three PDEGMA blocks were different. PDMAEMA block quaternization introduced a positive charge; then, the block copolymer was adsorbed onto the negatively charged nanocellulose dispersed in water. It was also shown that a polyelectrolyte was present in nanocellulose. The modified nanocellulose exhibited thermally responsive behavior in solution upon heating and cooling, indicating that the properties of the polyelectrolyte could be transferred to cellulose. Non-electrolyte adsorption involves the physical binding of cellulose to a non-electrolyte polymer. Examples of cellulose surface-adsorption copolymers have been reviewed, some of which have a better ability to bind cellulose [58] (Figure 4).

#### 3.2.2. Chemical Modification Method

The chemical modification of cellulose occurs mainly through a chemical reaction involving -OH of the cellulose molecular chain and compound esterification, an etherification reaction, to produce cellulose ethers and other derivatives. The surface properties of cellulose fibers can be modified by chemical reactions on the surfaces of the fibers, which can introduce small molecular groups (polar or non-polar) or polymers. The active sites where chemical reactions occur are generally the hydroxyl groups of cellulose fibers or the functional groups generated before or during cellulose fiber pretreatment. Cellulose fiber pretreatment is primarily used to reduce energy consumption, whereas cellulose fiber surface modification is used to improve the compatibility or dispersion between cellulose and other substances.

However, cellulose modification is prone to carboxylation. The 2,2,6,6-tetramethylpiperidin-1-oxyl radical (TEMPO) oxidizer is a pretreatment agent that promotes nanofiber separation by selectively introducing carboxyl (acidic) groups at the C6 position of the glucose unit [59]. This method was used as a pretreatment to enhance the mechanical decomposition of cellulose, during which cellulose secondary alcohols were first oxidized to aldehyde groups by sodium iodate and then converted to carboxyl groups with sodium chlorite during the reaction. The degree of nanofibrillation of hardwood cellulose pulp was improved via homogenization using a periodate-chlorite continuous-zone selective oxygenation method. When the carboxyl group content in the oxidized cellulose was in the range of 0.38–1.75 mmol/g, the nanofibers formed high-viscosity transparent gels with a yield of 85–100% without blocking the homogenizer. Based on the field emission scanning electron microscopy images, the typical width of the obtained nanofibers was approximately 25 ± 6 nm. Based on the wide-angle X-ray diffraction results, all nanofiber samples maintained the crystalline structure of cellulose I, with a crystallinity index of approximately 40% [60].

Cellulose can be esterified with organic acids, acyl halides, acid anhydrides, and inorganic acids to produce mono-, di-, and tri-substituted cellulose esters, but from the point of view of process and industrial applications, the most important of these is cellulose nitrate. The effects of the mixed acid composition, mass ratio, nitration temperature, and time on the properties and yield of cellulose nitrate prepared from unconventional raw materials, such as oat hulls from large-tonnage grain processing residues, were investigated. Cellulose nitrates were prepared under optimal synthesis conditions with 98% solubility in alcohol–ether mixtures [61].

#### 3.2.3. Comparison of Different Modification Methods

The physical method of cellulose pretreatment is simple, convenient, and easy to operate, but the performance of modified products is unstable and the modifier easily falls off from the cellulose, resulting in a reduction in product performance. The chemical method is a better modification method, and it results in other properties of cellulose without changing its performance. Regarding the advantages and disadvantages of each cellulose modification method, a detailed comparison of the physical and chemical modification methods is presented in Table 2, which provides a theoretical basis for future modification methods for different samples. With improvements in the processing technology of cellulose materials, it will be possible to gradually replace traditional petroleum products to alleviate energy and environmental pressures as needed in the future.

## 4. Hemicellulose

Hemicelluloses represent promising renewable biomass as plant cell wall polysaccharides synthesized by glycosyltransferases on Golgi cell membranes and biopolymers; they are second only to cellulose in plant fibers. The hexose in hemicellulose is used in fermentation to produce fuel alcohol [62]. It also results in a reduction in sorbitol production [63]. Further, hemicellulose has been produced using xylose [64], xylitol [65], furfural [66], and feed yeast [67].

### 4.1. Separation and Extraction Methods

#### 4.1.1. Alkali Treatment Method

The solid residue obtained from hydrothermal pretreatment was loaded into the reactor with 100 g of pretreated corn stover fibers and 900 mL of deionized water containing NaOH after pretreatment. After 2 h, the insoluble residue was recovered via filtration, washed to neutrality, and dried to a constant weight at 45 °C. The pH of the combined filtrates was adjusted to 5.0 using 6 mol of HCl. Three times the volume of 95% ethanol (*v*/*v*) was then added to precipitate hemicellulose for 2 h [43]. After filtration, the solid residue was freeze-dried to obtain the hemicellulose. Sweet sorghum stem hemicellulose can be extracted using similar methods. In one study, sweet sorghum stems were soaked in a 2.0% KOH aqueous solution at 90 °C for 3 h of continuous extraction with water; solids were filtered through a Bronsted funnel; and 60% ethanol was used for precipitation to obtain hemicellulose with a content of 76.3% [45]. In the ultrasonically-assisted alkaline extraction of hemicellulose from sugarcane bagasse pith, the total hemicellulose yield was up to 23.05% under the optimal conditions of an ultrasonic treatment time of 28 min, a KOH mass concentration of 3.7%, and an extraction temperature of 53 °C. The total hemicellulose yield was significantly increased by 3.24% compared with the case without ultrasonically-assisted extraction [68]. Alkaline peroxides can cause hemicellulose to be released into the sample under certain conditions, and this treatment does not change the overall structure of the hemicellulose. For the isolation and purification of hemicellulose polysaccharides from the dietary fiber of okara, which is rich in dietary fiber, the method comprised mixing the sample with 21.2 mL of 10% NaOH after pre-treatment at 35.5 °C and extraction for 5.3 h. Under these conditions, the yield of polysaccharides reached 30.21% [69]. At present, the separation and extraction of hemicellulose from insoluble dietary fiber has a low extraction rate and low quality, and cannot meet industry demands.

#### 4.1.2. Separation and Extraction of Organic Solvents

Unlike the alkali treatment method, organic solvent separation and extraction can separate and extract hemicellulose without pretreatment or the separation of cellulose and lignin. Organic solvents can effectively prevent the removal of acetyl groups from the hemicellulose functional groups in plant cells, resulting in high purity and proximity to the original hemicellulose structure [70]. Currently, dimethyl sulfoxide is the organic solvent used for the separation and extraction of hemicellulose. The effect of the aqueous extraction of hemicellulose from wheat straw using a mixture of formic acid, acetic acid, and ethanol on its content was investigated. Formic acid–acetic acid–H_2_O was the best system, producing 76.5% of the original hemicellulose from wheat straw. Moreover, organic solvent extraction is the most convenient and produces the highest hemicellulose content among all available separation and extraction methods [49]. In this study, the hemicellulose content extracted from barley straw and corn stalk cell walls using organic solvent extraction was visualized. Aqueous solutions of 90% neutral dioxane, 80% dioxane (containing 0.05 mol HCl), dimethyl sulfoxide, and 8% KOH were used to successfully extract 94.6% and 96.4% of the original hemicellulose from barley straw and corn stalks, respectively [71].

#### 4.1.3. Basic Hydrogen Peroxide Extraction Method

Alkaline hydrogen peroxide extraction is a common method for separating plant hemicellulose. Hydrogen peroxide, under alkaline conditions, not only has a removal and bleaching effect on lignin, but also improves the solubility of large-molecular-size hemicelluloses [72]. Thus, it can be used as a mild hemicellulose solubilizer. A comparative study of two methods for the separation of hemicellulose from rice straw using alkali extraction and alkaline hydrogen peroxide showed that 67.2% hemicellulose could be obtained with alkali extraction alone, and the addition of different concentrations of hydrogen peroxide increased the hemicellulose content to 88.5%. It also had a whiter color [73]. The optimal conditions for the extraction of bagasse hemicellulose from alkaline hydrogen peroxide solution were investigated in depth, and the best reaction conditions were determined to be 6% H_2_O_2_ mass fraction, 4 h reaction time, 20°C reaction temperature, and 0.5% magnesium sulfate addition, under which the hemicellulose content reached 86% and the product contained very little conjugated lignin (only 5.9%) [52].

#### 4.1.4. Comparison of Different Isolation and Extraction Methods

Regarding the advantages and disadvantages of each hemicellulose separation and extraction method, a detailed comparison of the three methods (alkali treatment, organic solvent extraction, and alkaline hydrogen peroxide method) is presented in Table 3 to provide a theoretical basis for the future separation and extraction of hemicellulose, as well as for the development of insoluble dietary fiber hemicellulose from okara.

### 4.2. Modification

Unmodified hemicellulose cannot be fully utilized because of the complexity of its structure. To explore the potential applications of hemicellulose, its modifications have been investigated both domestically and internationally. Because of the presence of numerous hydroxyl groups in both the main and side chains of hemicellulose [74], semifibers can be modified by oxidation, esterification [75], etherification [76,77], and grafting copolymerization [78].

Oxidation is the process of converting alcohol hydroxyl groups of hemicellulose into aldehyde or carboxyl groups. Oxidative modification can create carboxylic acids, the further reactive modification of which can solve the problem of the poor stability of hemicellulose-based materials. Carbonylated anionic galacturonic acid derivatives were prepared using a combined biological enzyme-oxidation reaction system [79]. When hemicellulose is combined with hydrophobic materials as a raw material, the hydrophilic property limits its interfacial integration with the resin, which in turn affects the mechanical properties of the synthesized product [80], and the esterified hemicellulose can solve the problem of excessive hydrophilicity [81]. The hydrophilic properties and polysaccharide attributes of hemicellulose make it advantageous for food preservation, microbial culture, and biopharmaceuticals. However, the excessive hydrolytic properties of hemicellulose under conditions of high humidity also limit its application, especially as a biopharmaceutical membrane material which requires good stability and surface activity; therefore, modification via etherification is necessary [82]. Copolymerization modification can also cause hemicellulose to acquire the properties of some grafting groups: i.e., grafting halogen groups to improve flame retardancy; grafting hydroxyl and aldehyde groups to improve hydrophilicity; and grafting acyl groups to increase the hydrophobicity of the material [83].

#### 4.2.1. Etherification Modification

One of the most common methods for carboxymethylation in etherification reactions involves changing the properties of hemicellulose by introducing a carboxymethyl group into its hydroxyl group. In the case of konjac glucomannan, the hydrogen in the glucomannan molecule is replaced by a carboxymethyl group (etherification) during the reaction with chloroacetic acid in a sodium hydroxide solution, resulting in carboxymethyl glucomannan [84]. For the preparation of co-blended membranes from quaternate hemicellulose (QH) and carboxymethyl cellulose (CMC), the QH and CMC solutions were first mixed to form a homogeneous suspension and then dried under a vacuum to prepare the hybrid film. From the results of the mechanical properties and water vapor permeability (WVP), the blended film exhibited good tensile strength and transmittance and low WVP for applications in coatings and packaging [85].

#### 4.2.2. Transesterification Modification

In addition to etherification, the hemicellulose esterification reaction results in new functions for hemicellulose, with its advantages including water resistance, hydrophilicity, thermal stability, and surface activity. Hemicellulose can be esterified using a variety of compounds, such as sulfuric acid reagents, chloride, and acid anhydride. The sulfation of hemicellulose is a process in which the hydroxyl group of hemicellulose reacts with the sulfonic acid group to dehydrate it. Xylan sulfate was obtained from alkali-soluble bagasse via sulfation with chlorosulfonic acid and N,N-dimethylformamide. Previously, a product with a degree of substitution of 1.49 and a molecular weight of up to 148,217 could be obtained in a flow system, even at room temperature, within 10 min [86].

#### 4.2.3. Comparison of Different Modification Methods

The advantages and disadvantages of hemicellulose modification methods for both etherification and esterification reactions are listed in Table 4. With the development of technology, the extraction of hemicellulose will become increasingly easier, and there will be an increasing number of modification methods. Moreover, the modified hemicellulose can be widely used in various industries, which will bring great social and economic benefits.

## 5. Lignin

### 5.1. Separation and Extraction Methods

Lignin separation and extraction are prerequisites which are important for the high-value utilization of lignin. Lignin has a complex molecular structure, containing crosslinked polymers of phenolic monomers, particularly p-coumaryl alcohol, coniferyl alcohol, and sinapyl alcohol [36] (Figure 5). Lignin can be used for different industrial and biomedical applications, including chemical substances, polymers, biofuels, and drug delivery, which are applications for the development of nano-materials [87]. Currently, lignin is mainly used as a binder, dispersant, chelating agent, stabilizer, emulsifier, and composite material, but associated research is still limited. Previously, the market share was low [88]. However, recently, the colloidal nature of lignin has attracted widespread attention for industrial applications, and the preparation of lignin for stabilizing emulsions [89] and the delivery of hydrophobic molecules shows promise as an alternative to toxic nanoparticles [90]. Based on this, several lignin extraction methods have been discussed, including DES, organic acid extraction, and ionic liquid (IL) treatment.

#### 5.1.1. DES Method

The DES method is characterized by the formation of a homogeneous and clarified solvent mixture by heating a hydrogen bond acceptor, choline chloride, with a different hydrogen bond donor while stirring at higher temperatures [91]. When a dietary fiber component is separated and extracted, the hydrogen bond donor is replaced to achieve the desired result. DESs have become promising for lignocellulosic biomass fractionation because of their high selectivity and environmentally friendly nature [92]. ChCl and LA were mixed in a sealed glass vial at 60 °C at a 1:2 molar ratio in a vacuum oven for 2 h, with regular stirring, until a homogeneous and clear liquid was obtained. Lignin was not consistently separated during pretreatment with hot water. A lignin content of 30.97% was successfully extracted from red winter wheat straw using a synergistic DES solvent-assisted hot water synergistic treatment. However, the complex interlocking structures of cellulose, hemicellulose, and lignin and the unique properties of lignin limit its value-added utilization [93]. The preparation of DES has been achieved from choline chloride-lactic acid (ChCl-LA) to extract lignin nanoparticles from herbal biomass (wheat straw) [94] (Figure 6). Further, the DES was found to be able to extract high-purity lignin (up to 94.8%) from wheat straw.

#### 5.1.2. Organic Acid Processing

The difficulty of isolating lignin is attributed to its complex structure, including non-hydrolyzable monomers and isomerism. Organic acid treatment is one of the most effective methods for separating lignin, and it promotes the degradation of carbohydrates during the treatment process [95]. Lignin can also be extracted from sugarcane bagasse using a phosphorylation solution. Bagasse was placed in a conical flask and heated in a water bath. The solid–liquid ratio was determined to be 1:20, and the reaction was carried out at 80 °C for 20 min using a p-TSOH solution with a concentration of 80%. Further, sugarcane bagasse achieved 88.81% lignin removal after phosphorylation [96]. Different preparation conditions are used to separate lignin from biomass depending on the production needs, as well as the production purpose. When the pretreatment effect is unsatisfactory, an organic acid treatment is used. The separation of cellulose fibers and lignin from red hemp bast using microwave-assisted organic acid treatment has also been studied [97]. Red hemp bast (12 g) was placed into four different solvents, including lactic acid, formic acid, acetic-acid, and a formic acid/acetic acid/water mixture, with a liquid–solid ratio of 20:1 and a removal rate of 94.68% after heating and stirring at 130 °C for 30 min [98].

#### 5.1.3. IL Treatment Method

ILs are organic salts composed of organic cations or anions in a liquid state at room temperature, and can also be called room-temperature ILs [99]. These represent a new type of solvent with the advantages of almost no vapor pressure, non-flammability, non-volatility, and good chemical stability and recyclability. This is thus referred to as a “green chemical solvent”, which can also be used as an alternative to low eutectic solvents. Preliminarily, it was shown that IL pretreatment could effectively disrupt the macromolecular structure of lignin and achieve its initial depolymerization. Lignins comprise various aromatic and phenolic compounds. In one study, an ionic (H_2_P_2_0_4_−) solution was mixed with wheat straw and rice husks, which dissolved 73% of the lignin at 100 °C for 2 h [100].

#### 5.1.4. Comparison of Different Methods

The advantages and disadvantages of different lignin extraction methods are shown in Table 5.

### 5.2. Modification Technology

Lignin contains many active functional groups, including carbonyl, methoxy, and hydroxyl groups. Several chemical modification methods for lignin have been discussed, including etherification and graft co-polymerization. Lignin can also be used as an antioxidant [101,102] and flame retardant [103]. Combined with polymer materials, this can reduce the production cost of polymer materials and the plasticity and fluidity of the products, thus increasing the performance and adding value to the products.

#### 5.2.1. Etherification Modification

The most commonly used etherification method involves the modification of propylene oxide in an alkaline solution to prepare lignin-based epoxy resins. The resulting solution is treated with epichlorohydrin and cured via crosslinking with m-phenylenediamine [104]. Etherification reactions can produce new polyols, and are among the most promising modification methods available [105]. Insoluble lignin and other solids can be converted into water-soluble polyols through treatment with various organic solvents. This method has been used extensively for different biopolymers and bio-based materials containing hydroxyl groups, such as chitosan [106], corky [107], corn starch [108], and beet pulp [109,110]. This modification method allows for the extraction of a wide range of polyols from various biomass residues, which can be used to produce new polymeric materials, such as polyurethane foams [111]. The addition of lignin improves the mechanical and thermal properties of epoxy resins, which is attributed to the presence of aromatic groups in the lignin structure [112].

#### 5.2.2. Graft Co-Polymerization Modification

Graft co-polymerization is the formation of a chain bond between polymer B and polymer A, which can be expressed as A-graft-B or A-g-B. Graft copolymers improve the mechanical properties of composites, reduce friction, and decrease flammability. Insoluble dietary fiber is one of the main components of okara. A single fraction of insoluble dietary fiber isolated from okara was used to produce graft polymers. This was performed by mixing okara with water in a 75% suspension, followed by homogenization. This suspension was placed in a 250 mL triangular flask equipped with a stirrer and a nitrogen line, and the suspension was treated with nitrogen for 15 min and then heated to 70 °C for 15 min under a stream of nitrogen. After adding the initiator APS (144 g) and maintaining it at 70 °C for 30 min under a stream of N_2_, a solution was prepared by adding 7.2 g of AA to 16.6 mL of water. The reaction mixture was maintained under N_2_ at 70 °C for 5 h. After graft polymerization was complete, the reaction mixture became a viscous product called Ok-PAA. Then, 10.3 g of the resulting viscous product was suspended in deionized water and centrifuged at 11,000 rpm for 20 min. The precipitate was collected, washed with water, centrifuged for 3 h, freeze-dried, and named OK-PAA (pre) (yield: 0.588 g). The supernatants were collected, combined, freeze-dried, and named Ok-PAA (sup) (yield: 0.815 g) [113] (Figure 7).

Graft polymerization involves the addition of different initiators to the polymer to be grafted, and the final reaction product after graft polymerization is obtained by heating and stirring the mixture for a certain period. Lignin’s co-polymerization with vinyl improves the reactivity of lignin and generates new graft copolymers. Previously, the polymerization reaction was initiated with ferrous chloride and hydrogen peroxide, and the grafting efficiency was maintained at approximately 18%, independent of the mass fraction of the initiator [114]. The grafting efficiency of methyl methacrylate with lignin was reduced by side reactions with phenolic hydroxyl groups when free-radical initiators were used [115]. This method provides a foundation for the preparation of lignin–graft co-polymers.

#### 5.2.3. Comparison of Different Modification Methods

Etherification and graft co-polymerization are the most commonly used methods for modifying lignin. The advantages and disadvantages of both methods are listed in Table 6. However, this chemical modification method is difficult to use for the development of edible lignin. Therefore, further research on lignin will be of great significance for achieving green development.

## 6. Conclusions

In the current world of energy scarcity and severe environmental pollution, sustainable development is imperative. This paper reviews methods for the separation and extraction of cellulose, hemicellulose, and lignin from insoluble dietary fiber. Further, technology for the modification of single components in insoluble dietary fibers is discussed scientifically, allowing for a detailed understanding of this topic.

Although this review describes a variety of methods for the isolation and extraction of insoluble dietary fiber monomers, the most commonly used methods involve sugarcane and bamboo. However, these gramineous plants are not as rich in dietary fiber. For example, sugarcane contains only 0.60 g of dietary fiber per 100 g. The soybean consumption level in China is among the highest in the world, and most okara is currently discarded as feed or waste and is not fully and reasonably utilized. Many resources are wasted, which also pollutes the environment. Currently, there are few methods or theoretical bases for separating and extracting single fractions of insoluble dietary fiber from okara, making this the foundation for future development. For example, cellulose in insoluble dietary fiber from soybean residue is a renewable biomass resource that can be converted into biofuels such as biogas, bioethanol, and biodiesel through biomass energy conversion technology in the near future, which may be used as a secondary energy source. In addition, cellulose can also be converted into energy sources such as biohydrogen and biomethane through biomass fermentation technology, and can also be used to produce chemicals and materials. Therefore, the application of okara-insoluble dietary fiber to separate and extract single components has improved in various industries. For example, it can be used in food packaging bags, nano-materials, and sensors. This application also improves the utilization value of okara-insoluble dietary fiber. Okara-insoluble dietary fiber can also be used to produce rubber and bioindicators using modified technology. The use of okara-insoluble dietary fiber represents the rational use of byproducts to meet the need to conserve resources, as well as for sustainable human development.

## Figures and Tables

**Figure 2 foods-12-02473-f002:**
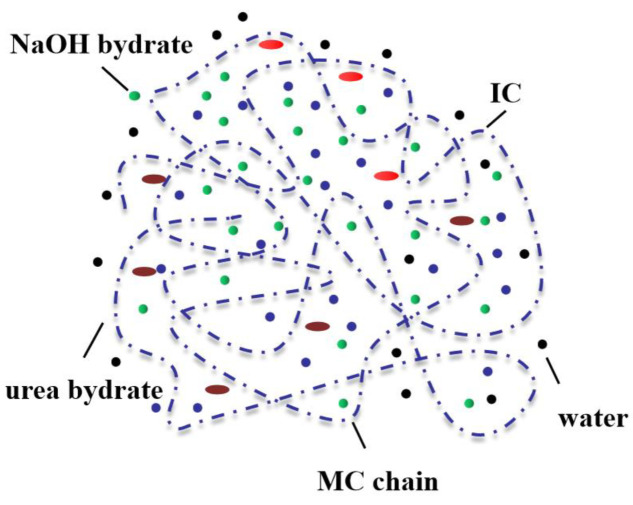
Interaction between NaOH/urea and cellulose [44].

**Figure 3 foods-12-02473-f003:**
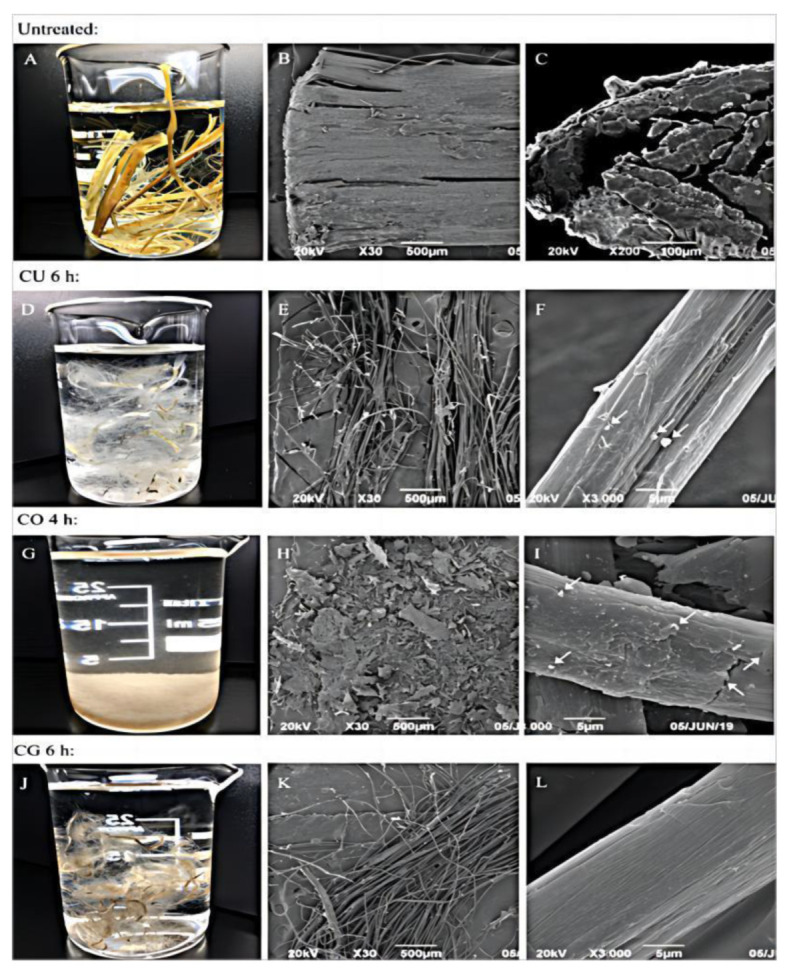
Deep eutectic solvent (DES) treatment of ramie raw fiber [50]. (**A**–**L**) Digital photographs and SEM images of raw RFs and pretreated RFs.

**Figure 4 foods-12-02473-f004:**
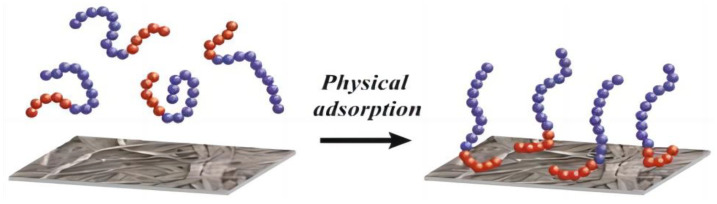
Physical adsorption [58].

**Figure 5 foods-12-02473-f005:**
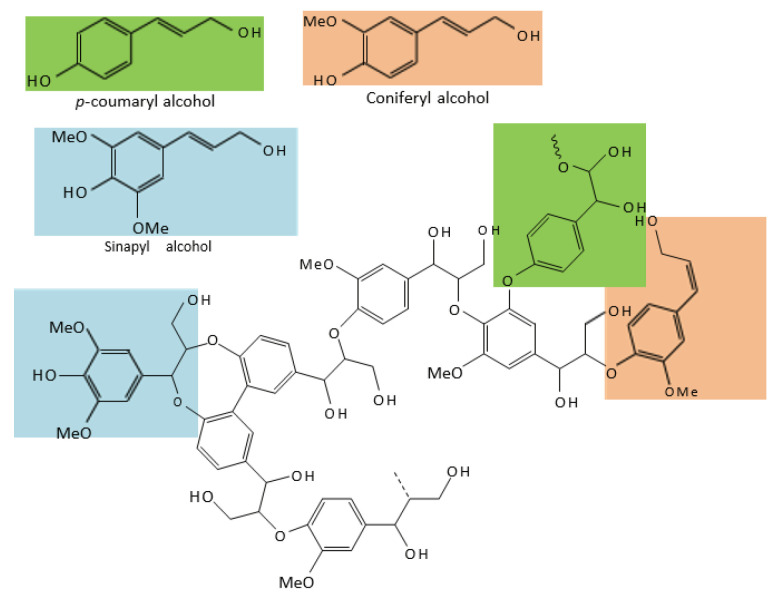
Chemical structures of lignin (p-coumaryl alcohol, coniferyl alcohol, and sinapyl alcohol) [36].

**Figure 6 foods-12-02473-f006:**
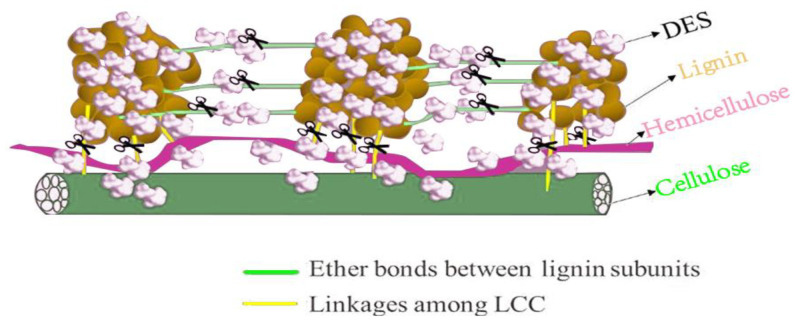
Link between deep eutectic solvent (DES) and single components [95].

**Figure 7 foods-12-02473-f007:**
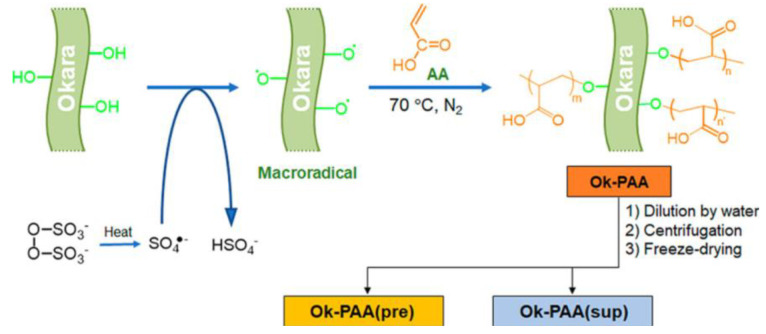
Synthesis and post-treatment procedures for Ok−PAA graft polymers [113].

**Table 1 foods-12-02473-t001:** Comparison of different methods to separate single components from fiber.

Extraction Methods	Advantages	Disadvantages
Acid hydrolysis method	(1) Simple process (2) Easy operation and no danger (3) Cellulose becomes nano-sized (4) Cellulose has high thermal (5) Stability and uniform particle size	(1) Large quantities of acid and impurities remain in the reactants (2) Difficult to recover and can cause environmental pollution (3) Can cause damage to the structure of cellulose
Steam blasting treatment	(1) Low energy input (2) Does not require recycling reagents (3) Little impact on the environment (4) Adding acid or bases improves the treatment efficiency	(1) Prone to Merad reaction under high temperature and pressure (2) Target products are readily degradable (3) Sample volume increases
Deep eutectic solvent method	(1) Simple preparation (2) Recyclable (3) Method (4) Non-toxic and degradable (5) Consistent with the concept of sustainability	(1) Easily soluble in water and in large amounts

**Table 2 foods-12-02473-t002:** Comparison of different modification methods.

Extraction Methods	Advantages	Disadvantages
Physical modification method	Simple and convenient pre-processing, easy to operate	(1) Unstable product performance (2) Modifier easily comes off from the cellulose, resulting in a decrease in product performance
Chemical modification method	Imparting other properties to cellulose without changing its properties	(1) Pollution of the environment (2) Has reagent residue

**Table 3 foods-12-02473-t003:** Comparison of different isolation and extraction methods.

Extraction Methods	Advantages	Disadvantages
Alkali treatment method	(1) Low cost (2) Pure and non-polluting products	(1) Stronger bases generate new amino acids
Organic solvent extraction method	(1) Low cost (2) Pure and non-polluting products (3) Removal of lignin and bleaching	(1) Peroxide has strong oxidizing properties and the possibility of combustion
Alkaline hydrogen peroxide method	(1) Direct and effective (2) Closest to the original structure	(1) High energy efficiency consumption (2) Some chemical reagents will produce precipitation

**Table 4 foods-12-02473-t004:** Comparison of different modification methods.

Extraction Methods	Advantages	Disadvantages
Etherification modification	(1) Strong cationic properties (2) Strongly water-soluble	(1) Decrease in water content (2) Increase in hydrophobicity
Esterification modification	(1) Results in new properties for hemicellulose (2) Enhanced water resistance and hydrophilicity (3) Increased thermal stability and surface activity	(1) Reaction is reversible

**Table 5 foods-12-02473-t005:** Comparison of different lignin extraction methods.

Extraction Methods	Advantages	Disadvantages
DES method	See Section 3.1.4 for details	See Section 3.1.4 for details
Organic acid extraction	(1) Obtains lignin quickly (2) High-purity lignin is obtained	(1) Organic solvents are not easily recovered (2) Pollutes the environment
Ionic liquid method	Similar to DES method	Similar to DES method

**Table 6 foods-12-02473-t006:** Comparison of different modification methods.

Extraction Methods	Advantages	Disadvantages
Etherification modification	Better dissolution of lignin	(1) Not friendly to the environment (2) Highly polluting
Graft co-polymerization method	Increased lignin reactivity	(1) Most initiators are toxic

## Data Availability

Raw data can be provided by the corresponding author upon request.

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
