# Peer review of "Isolation and Extraction of Monomers from Insoluble Dietary Fiber"

_foods, 2023, doi:10.3390/foods12132473_

Round 1

Reviewer 1 Report (New Reviewer)

Dear Authors,

This is excellent work, and with some minor revisions, this work has the potential to be even more impactful. Please consider the following:

1.      Line 12-13. This sentence is out of the context of the paper. What is the relationship between fiber and energy?

2.      The introduction can be enhanced by the inclusion of a recent study on the subject.

3.      Please improve the quality of Figure 4.

4.      Figure 6 could be improved if the background is removed.

Author Response

Dear Reviewer: 

Thank you for your letter and the reviewers' comments concerning our manuscript entitled “Isolation and Extraction of Monomers from Insoluble Dietary Fiber”(ID: foods—2440454) submitted for publication in FOODS.Those comments are all valuable and very helpful for revising and improving our manuscript, as well as the important guiding significance to our research.

We studied the comments carefully and revised the manuscript accordingly to meet the editorial and reviewer’s satisfaction and approval. Revised portions are marked Red.

  1. Line 12-13. This sentence is out of the context of the paper. What is the relationship between fiber and energy?The introduction can be enhanced by the inclusion of a recent study on the subject.

Response: Thanks for the reviewer good suggestion. We have revised the manuscript as follows and have also added a note on this topic in the introduction section.

(Page 1, lines 12-13)

Previous:With increasing awareness of the increasing scarcity of non-renewable energy sources, the conversion of non-renewable energy into renewable energy and its use have become a constant concern.

Corrective:With the recognition of the increasing scarcity of non-renewable energy sources, the conversion of single components of dietary fiber into renewable energy sources and their use has become an ongoing concern.

(Page 2, lines 51-54)

Corrective:Developing economically viable and sustainable technologies for converting corn fiber into liquid fuels is widely seen as a promising approach to achieve improved ethanol titers and integrated utilization of byproducts in the traditional corn dry milling process [15]. 

  1. Please improve the quality of Figure 4.

Response: Thanks for the reviewer good suggestion. We have revised Figure 4 in higher resolution.

Corrective:

Note: Since the changed image cannot be displayed here, we have uploaded the full response as a file.

Figure 4. Physical adsorption [59].

  1. Figure 6 could be improved if the background is removed.

Previous:

Note: Since the changed image cannot be displayed here, we have uploaded the full response as a file.

Figure 6. Link between deep eutectic solvent (DES) and single components [97].

Corrective:

Note: Since the changed image cannot be displayed here, we have uploaded the full response as a file.

Figure 6. Link between deep eutectic solvent (DES) and single components [97].

We seriously appreciate the enthusiastic work of our reviewers and hope that the revisions will be approved. Thank you again for your good comments and suggestions.

Sincerely,

Hansong Yu, Wendan Jing

College of Food Science and Engineering, Jilin Agricultural University, Changchun 130118, China;

+86 133-3176-0468, +86 188-4419-1415

yuhansong@163.com, jwddoc@163.com

Reviewer 2 Report (New Reviewer)

Comparison of various methods was very  good represented in the Tables with brief description of advantages and disadvantages.

Future perspectives must be described specifically

The provided information in manuscript is useful though it would be beneficial to  additionally cite and discuss the relevant publications such as

Hua, M., Lu, J., Qu, D., Liu, C., Zhang, L., Li, S., ... & Sun, Y. (2019). Structure, physicochemical properties and adsorption function of insoluble dietary fiber from ginseng residue: A potential functional ingredient. Food Chemistry286, 522-529.

Xie, Yitong, et al. "Efficient extraction and structural characterization of hemicellulose from sugarcane bagasse pith." Polymers 12.3 (2020): 608.

Huang, Ling-Zhi, et al. "Recent developments and applications of hemicellulose from wheat straw: A review." Frontiers in Bioengineering and Biotechnology 9 (2021): 690773.

Zhang, X., Zeng, Y., Liu, J., Men, Y., & Sun, Y. (2023). Effects of three extraction methods on the structural and functional properties of insoluble dietary fibers from mycoprotein. Food Chemistry Advances, 100299.

The English is understandable

Author Response

Dear Reviewer:

Thank you for your letter and the reviewers' comments concerning our manuscript entitled “Isolation and Extraction of Monomers from Insoluble Dietary Fiber”(ID: foods—2440454) submitted for publication in FOODS.Those comments are all valuable and very helpful for revising and improving our manuscript, as well as the important guiding significance to our research.

We studied the comments carefully and revised the manuscript accordingly to meet the editorial and reviewer’s satisfaction and approval. Revised portions are marked Red.

  1. Future perspectives must be described specifically.

Response: Thanks for the reviewer good suggestion. We have made the following revisions to the manuscript.

(Page 16, lines 631-637)

Corrective:For example, cellulose in insoluble dietary fiber from soybean residue is a renewable biomass resource that can be converted into biofuels such as biogas, bioethanol and biodiesel through biomass energy conversion technology in the near future, which can be used as a secondary energy source. In addition, cellulose can also be converted into energy sources such as biohydrogen and biomethane through biomass fermentation technology, and can also be used to produce chemicals and materials.

  1. The provided information in manuscript is useful though it would be beneficial to additionally cite and discuss the relevant publications.

Response: Thanks for the reviewer good suggestion. We will cite four documents that contributed to the article into the manuscript.

(Page 1, lines 47) and (Page 2, lines 48-50)

Corrective:Ginseng-IDF has a typical hydrolyzed fiber structure, polysaccharide functional groups and cellulose crystal structure, which can also be used as an ideal functional ingredient for food processing [13].

(Page 9, lines 346-351)

Corrective:In the ultrasonic-assisted alkaline extraction of hemicellulose from sugarcane bagasse pith, the total hemicellulose yield was up to 23.05% under the optimal conditions of ultrasonic treatment time of 28 min, KOH mass concentration of 3.7%, and extraction temperature of 53°C. The total hemicellulose yield was significantly increased by 3.24% compared with the case without ultrasonic-assisted extraction[70].

(Page 3, lines 113-116)

Previous:Hemicellulose, one of the main components of insoluble dietary fiber, is abundant, biodegradable, renewable, and biocompatible in nature. Moreover, it is widely used in many fields such as food packaging materials [30], paper coatings [31], chemicals, environmental energy, and biomedicine.

Corrective:Hemicellulose, one of the main components of insoluble dietary fiber, is abundant, biodegradable, renewable, and biocompatible in nature. Moreover, it is widely used in many fields such as food packaging materials [30], paper coatings [31], chemicals[32], environmental energy, and biomedicine.

(Page 1, lines 44-45)

Corrective:IDF is important as a functional food ingredient with multiple health and nutritional benefits [10].

References:

[13] Hua, M., Lu, J., Qu, D., Liu, C., Zhang, L., Li, S., ... & Sun, Y. (2019). Structure, physicochemical properties and adsorption function of insoluble dietary fiber from ginseng residue: A potential functional ingredient. Food Chemistry, 286, 522-529.

[70] Xie, Yi tong, et al. "Efficient extraction and structural characterization of hemicellulose from sugarcane bagasse pith." Polymers 12.3 (2020): 608.

[32] Huang, Ling-Zhi, et al. "Recent developments and applications of hemicellulose from wheat straw: A review." Frontiers in Bioengineering and Biotechnology 9 (2021): 690773.

[10] Zhang, X., Zeng, Y., Liu, J., Men, Y., & Sun, Y. (2023). Effects of three extraction methods on the structural and functional properties of insoluble dietary fibers from mycoprotein. Food Chemistry Advances, 100299.

We appreciate for Reviewers’ warm work earnestly, and hope that the corrections will meet with approval. We are looking forward to your information about our revised manuscript and thanks again for your good comments and suggestions.

Sincerely,

Hansong Yu, Wendan Jing

College of Food Science and Engineering, Jilin Agricultural University, Changchun 130118, China;

+86 133-3176-0468, +86 188-4419-1415

yuhansong@163.com, jwddoc@163.com

This manuscript is a resubmission of an earlier submission. The following is a list of the peer review reports and author responses from that submission.

Round 1

Reviewer 1 Report

Dear Authors,

Several mistakes were found in your manuscript. Several of these where highlighted in your manuscript. These most be corrected by the authors.

Sincerely yours,

The Reviewer
